Self-assembly of Hyaluronic Acid-Cu-Quercetin flavonoid nanoparticles: synergistic chemotherapy to target tumors

Yue Hanxun 1 2
Zhao Xuan 1
Yong Qin 1
Shi Min 1
Jiang Xiaofeng 1
Zhang Yating 1
Yu Xian 1 303671@cqmu.edu.cn
1 Phase I Clinical Trial Center, The Second Affiliated Hospital of Chongqing Medical University , Chongqing , China
2 The First people’s Hospital of PingDingShan , Pingdingshan , China
Sotelo-Mundo Rogerio
Electronic publication date: 2023 Aug 28
Publication date: 2023
Volume: 11
Electronic Location ID: e15942
Received 2023 Mar 7; Accepted 2023 Jul 31
Copyright: © 2023 Yue et al.
Copyright year: 2023
Copyright holder: Yue et al.
License: This is an open access article distributed under the terms of the Creative Commons Attribution License, which permits unrestricted use, distribution, reproduction and adaptation in any medium and for any purpose provided that it is properly attributed. For attribution, the original author(s), title, publication source (PeerJ) and either DOI or URL of the article must be cited.
License URL: https://creativecommons.org/licenses/by/4.0/

Keywords: Quercetin, Quercetin-copper complex, Flavonoid self-assembled nanoparticles, Fenton-like reaction, Synergistic antitumor

Funding: National Natural Science Foundation of China 82072327 Chongqing Natural Science Foundation CSTB2022NSCQ-MSX0987 Kuanren Talents Program of the second affiliated hospital of Chongqing Medical University This work was financially supported by the National Natural Science Foundation of China (NO. 82072327), the Chongqing Natural Science Foundation CSTB2022NSCQ-MSX0987, and the Kuanren Talents Program of the second affiliated hospital of Chongqing Medical University. The funders had no role in study design, data collection and analysis, decision to publish, or preparation of the manuscript.

==============================
Background

In this study, a natural compound quercetin (Qu) was investigated for its various antitumor effects. However, due to its poor water solubility and low bioavailability, its clinical application is limited. To overcome this constraint, a modification was to Qu, which resulted in the creation of novel flavonoid self-assembling nanoparticles (HCQ NPs).

Methods

HCQ NPs were synthesized by a self-assembly method and characterized using transmission electron microscopy, the Malvern Zetasizer instrument, X-ray photoelectron spectroscopy (XPS), the ultraviolet-visible spectrophotometric method (UV-vis), Fourier transform infrared (FITR) and inductively coupled plasma mass spectrometry. Extracellular, methylene blue spectrophotometric analysis was used to determine the ability of HCQ NPs to react with different concentrations of H2O2 to form hydroxyl radicals (•OH). Intracellular, DCFH-DA staining was used to detect the ability of HCQ NPs to react with H2O2 to generate reactive oxygen species. Flow cytometry was used to detect the uptake of HCQ NPs by MDA-MB-231 cells at different time points. The biocompatibility of HCQ NPs was evaluated using the Cell Counting Kit-8 (CCK-8) assay. Calcein AM/PI double staining and the CCK-8 assay were used to evaluate the synergistic antitumor effect of HCQ NPs and H2O2.

Results

HCQ NPs showed uniformly sized analogous spherical shapes with a hydrodynamic diameter of 55.36 ± 0.27 nm. XPS revealed that Cu was mainly present as Cu2+ in the HCQ NPs. UV−vis absorption spectrum of the characteristic peak of HCQ NPs was located at 296 nm. Similarly, FTIR spectroscopy revealed a complex formation of Qu and Cu2+ that substantially changed the wavenumber of the 4-position C = O characteristic absorption peak. Based on the proportion of Qu and Cu2+ (1:2), the total drug loading of Qu and Cu2+ in the HCQ NPs for therapeutic purposes was calculated to be 9%. Methylene blue spectrophotometric analysis of •OH indicated that Cu can lead to the generation of •OH by triggering Fenton-like reactions. HCQ NPs rapidly accumulated in MDA-MB-231 cells with the extension of time, and the maximum accumulation concentration was reached at about 0.5 h. Calcein AM/PI double staining and CCK-8 revealed synergistic antitumor effects of HCQ NPs including the chemotherapeutic effect of Qu and chemodynamic therapy by Cu2+ in a simulated tumor microenvironment. HCQ NPs demonstrated very low toxicity in LO2 cells in the biocompatibility experiment.

Conclusion

This study show cases a new method of creating self-assembled flavonoid HCQ NPs that show great for fighting cancer.

Introduction

Cancer remains the primary cause of death globally. In 2020, the American Cancer Society documented 19.3 million fresh cases of cancer and 10 million cancer-related deaths (Sung et al., 2021). Combination chemotherapy using two or more drugs has been widely investigated in cancer therapy (Jhaveri, Deshpande & Torchilin, 2014). Although drugs with non-overlapping toxicity are selected as candidates for treatment, their combined treatment effects are often associated with higher toxicity compared to monotherapy (Tyagi et al., 2004; Zatloukal et al., 2003; Zhang et al., 2016). Additionally, the lack of targeting leads to an inability of the drugs to accumulate in the tumor tissues, limiting the synergistic effects of multiple drugs (Hua et al., 2018b; Lee, Saiful Yazan & Che Abdullah, 2017). Thus, currently used multidrug chemotherapies do not always yield expected outcomes. These problems could hypothetically be effectively overcome by encapsulating low-toxicity drugs in a targeted nanocarrier to form a nanodrug.

Many synthetic and natural compounds have been investigated for their potential antitumor effects (Sharifi-Rad et al., 2019). Many naturally occurring compounds have strong antitumor effects and excellent biocompatibility (Abdulridha et al., 2020; Hua, 2002). Quercetin (Qu) is a natural flavonoid compound widely present in daily diets and known for its antitumor effects (Rauf et al., 2018). In vivo and in vitro experiments have shown that Qu exerts its antitumor effects by altering the cell cycle, inhibiting cell proliferation, promoting apoptosis, and inhibiting angiogenesis (Pang et al., 2019). For example, in breast cancer MDA-MB-453 and MDA-MB-231 cells, Qu can halt the cell cycle at the G2/M phase by upregulating p53 (Chien et al., 2009; Tang et al., 2020). However, its clinical applications are limited due to low oral bioavailability, poor water solubility, rapid metabolism, and enzyme degradation (Scalia et al., 2013). Qu modification methods such as metal complex formation effectively overcome these challenges (Kakran et al., 2012). Hydroxyl and carbonyl groups present in Qu’s structure are very effective metal chelators (Leopoldini et al., 2006). The in vitro pharmacokinetic and biological activities of Qu-metal complexes are better than Qu alone due to their geometric spatial orientation and availability of metal ions at the binding site (Dolatabadi, 2011). For example, copper (Cu) ions can form a complex with Qu, demonstrating antitumor effects (Mutlu Gençkal et al., 2020). In addition, Cu ions can achieve synergistic antitumor effects by triggering a Fenton-like reaction to convert hydrogen peroxide (H2O2), which is excessively generated in the tumor microenvironment (TME), into more cytotoxic hydroxyl radicals (•OH) (Liu et al., 2019).

Hyaluronic acid (HA) is a highly efficient tumor-targeted delivery vehicle characterized by good biocompatibility, biodegradability, and unique CD44 receptor-binding capacity (Rippe, Cosenza & Auzély-Velty, 2019). The Qu-Cu complex can target tumor cells with high CD44 receptor expression by forming ionic bonds between HA and Cu ions. Therefore, we developed novel self-assembled multifunctional flavonoid nanoparticles (HCQ NPs) containing HA, CuCl2, and Qu. In these NPs, Qu serves as a self-loading chemotherapeutic agent, while Cu and HA act as Fenton-like reagents with respective targeted delivery. The fabricated NPs demonstrated good targeting and chemodynamic therapy (CDT) characteristics by effectively converting excess H2O2 in the TME into hydroxyl radicals to augment the antitumor effects of Qu synergistically.

Materials and Methods

Materials

Qu (95%), HA (98%), and dimethyl sulfoxide (DMSO) were purchased from Aladdin Industrial Co. (China). The H2O2 (30%) was purchased from Sinopharm Chemical Reagent Co., Ltd. Tris-HCl buffer (pH 8.8) and Cu (II) chloride dihydrate (CuCl2) were obtained from Sigma-Aldrich Co. (USA). Cell culture medium (RPMI-1640, complete DMEM high glucose), trypsin (for cell culturing, 0.25% w/v), and fetal bovine serum (FBS) were obtained from Thermo Fisher Scientific Co. Ltd. (China). The breast cancer cell line (MDA-MB-231) was provided by the School of Life Science and Technology, Chongqing Medical University, China. Human normal liver cells (LO2) were provided by the School of Life Science and Technology, Xiamen University, China.

Synthesis of HCQ NPs

The HCQ NPs were synthesized using a coordination-induced self-assembly method. Aqueous solutions of HA (10 mg/mL) and CuCl2 (5 mg/mL) were prepared. In addition, Qu solution (20 mg/mL) was prepared with DMSO. Then, 1 mL of the HA solution and 150 µL of the CuCl2 solution were added to a reaction flask and continuously stirred for 4 min while adding 150 µL of 1M Tris-HCl (pH 8.8). Subsequently, 125 µL of the Qu solution was added dropwise followed by stirring for 4 h at room temperature. The fabricated HCQ NPs were purified by dialysis overnight (MWCO: 10 kDa) and stored at room temperature until further experimentation.

Characterization of HCQ NPs

The fabricated HCQ NPs were characterized for various physical and chemical properties. Solution properties such as hydrodynamic diameters, zeta potentials, and stability of HCQ NPs were evaluated using a Malvern Zetasizer instrument (Nano-ZS90, UK) (n = 3). High-resolution transmission electron microscopy (TEM) (JEOL JEM-2100F, Japan) was used to characterize the morphological features of the NPs. The chemical structure of HCQ NPs was determined using Fourier transform infrared (FTIR) spectroscopy (Ettlingen, Germany). Ultraviolet-visible (UV-vis) absorption was collected using the Model Ultra-6600A (Rigol, China). X-ray photoelectron spectroscopy (XPS) was measured using ESCALAB250Xi (Thermo Fisher Scientific,Waltham, MA, USA) at 150 W. Inductively coupled plasma mass spectrometry (ICP-MS) was used to quantitatively measure the proportion of Cu in the samples at 1 KW transmission power with argon as the carrier gas. The drug loading content of the two drugs (Qu and Cu) in the HCQ NPs was calculated using the following formula (Hua et al., 2018a).

Drugloadingcontent(wt%)=massofthedruginnanoparticlesthetotalmassofthenanoparticles×100%

Extracellular chemodynamic activity of HCQ NPs

Different concentrations of H2O2 (1, 2, and 4 mM) were added to the HCQ NPs (200 μL, 5 mg/mL) and methylene blue (MB); (100 μL, 100 μg/mL), followed by stirring for 30 min at 37 °C in a dark vessel to protect from light. To evaluate the ability of HCQ NPs to produce hydroxyl radicals in vitro, the change in MB absorbance was measured at 666 nm using an UV-vis spectrophotometer.

Cell culture

Various cell lines including MDA-MB-231 and LO2 were incubated at 37 °C in RPMI-1640 medium containing 10% FBS and 1% antibiotics (penicillin−streptomycin, 10,000 U/mL) under 5% CO2.

Intracellular measurement of reactive oxygen species (ROS)

MDA-MB-231 cells (4 ×104) were seeded in a 24-well plate and cultured for 24 h. The PBS, HCQ, and HCQ + H2O2 groups were respectively incubated with 55 μL PBS, 50 μL HCQ NPs + 5 μL PBS, and 50 μL HCQ NPs + 5 μL H2O2 (final concentration of H2O2 was 50 μM) for 4 h. The cells were washed with PBS and incubated for 30 min in RPMI-1640 medium without FBS containing 10 mM of 2’,7’-dichlorofluorescein diacetate (DCFH-DA) in the dark. The intensity of green fluorescence between different groups was observed using an inverted fluorescence microscope (Nikon, USA) as an index of intracellular ROS.

Cellular uptake of HCQ NPs

The synthesized HCQ NPs were modified by an amide reaction with the fluorescent dye (cy-5) and purified to obtain cy-5-HCQ NPs. MDA-MB-231 cells (1 × 106) were seeded in 6-well plates and incubated for 24 h. Equal amounts of PBS or cy-5-HCQ NPs were added and incubated for different times (30 min, 2 h, 4 h, 8 h, 12 h, and 24 h). The cells were washed with PBS, trypsinized, and counted. The fluorescence intensity of each group was evaluated using flow cytometry (Beckman, Brea, CA, USA).

Cytotoxicity and synergism assays

The Cell Counting Kit-8 (CCK-8) assay was used to determine the cytotoxicity of HCQ NPs. MDA-MB-231 cells (3 ×103) were seeded into 96-well plates and incubated for 24 h. Different volumes of HCQ (5, 10, 15, and 20 μL), and HCQ (5, 10, 15, and 20 μL) + H2O2 (50 μM) (the volume difference of different groups supplemented with PBS) were added and incubated for 48 h. Different volumes of HCQ NPs (5, 10, 15, and 20 μL) represent various concentrations of HCQ NPs (Cu:Qu = 100 μM:50 μM, 190 μM:95 μM, 274 μM:137 μM, and 350 μM:175 μM) based on drug loading content of the two drugs (Qu and Cu) in the HCQ NPs. The 96-well plates were washed with PBS, and 100 μL RPMI-1640 containing 10% CCK-8 reagent was added to each well. The 96-well plates were placed in a CO2 incubator for 3 h. Relative cell viability was measured using a microplate reader (Thermo Fisher Scientific, Waltham, MA, USA) to detect the optical density (OD).

Calcein AM/PI double staining

MDA-MB-231 cells (5 × 105) were seeded into 6-well plates and incubated for 24 h. Cells were treated according to the grouping: PBS group (PBS 100 μL), HCQ group (HCQ 90 μL + 10 μL PBS), and HCQ + H2O2 group (HCQ 90 μL + 10 μL H2O2). Cell culture plates were placed in the incubator for 48 h. The cell suspension of each group was collected and mixed with AM/PI dye under dark conditions. Then, 150 μL cell suspension was added to a 24-well cell culture plate, and the cell culture plate was placed in a cell incubator for 30 min. The fluorescence in each group was observed under a fluorescent inverted microscope.

Biocompatibility assay

The CCK-8 assay was used to detect the biocompatibility of HCQ NPs. LO2 cells (3 × 103) were seeded into 96-well plates and incubated for 24 h. Different volumes of HCQ NPs (5, 10, 15, and 20 μL) were added and incubated for 48 h. The 96-well plates were washed with PBS, and 100 μL RPMI-1640 containing 10% CCK reagent was added to each well. The 96-well plates were placed in a CO2 incubator for 3 h. Relative cell viability was calculated using a microplate reader to detect the OD at 450 nm between different wells.

Results and discussion

Design and preparations

There have only been a few reports on synergistic outcomes of antitumor combinations involving the chemotherapeutic effects of natural antitumor drugs and CDT mediated by Fenton or Fenton-like ions (Fe2+, Cu2+). Since the molecular structure of naturally bioactive compounds mainly includes phenolic hydroxyl groups, double bonds, and keto groups, metallic ions can easily undergo coordinated reactions with these medicines (Jung et al., 2022). Naturally bioactive compounds also have antioxidant properties, which are contradictory to the action of Fenton or Fenton-like ions, which exert chemodynamic effects by producing ROS (Xue et al., 2018). For example, curcumin is a natural antitumor drug that has strong anti-oxidant properties, is capable of chelating with Fenton or Fenton-like ions (Fe2+, Cu2+), and has been used to treat Alzheimer’s disease because it inhibits ROS production (Chan et al., 2016; Zhai et al., 2015). However, other drugs such as gallic acid and gossypol can exert chemodynamic properties by reducing their coordinated Fenton or Fenton-like ions (Hao et al., 2020; Zaidi & Hadi, 1992). Qu is another naturally bioactive compound (Wu et al., 2019), and some studies have speculated that the antitumor effects of Qu-copper complexes are stronger than that of Qu alone because the complex can insert between DNA base pairs to inhibit DNA molecules (Li, Yang & Wu, 2010). These complexes can also produce highly toxic ROS through a Fenton reaction, thereby inducing oxidative DNA damage (Tan, Wang & Zhu, 2009). Although our pre-experimental results support this point, there is a lack of in vitro and in vivo validation experiments. In addition, antitumor nanodrug delivery systems have not been explored. Therefore, we constructed targeted self-assembled HCQ NPs with synergistic antitumor effects (chemotherapy and CDT) (Scheme 1) using Qu as a self-loading chemotherapeutic agent, Cu2+ as a Fenton-like reagent, and HA as a targeted delivery vehicle.

Scheme 1 Schematic illustration of the preparation of HCQ NPs as a versatile nanoplatform for efficient synergistic chemotherapy.

Characterization of HCQ NPs

The TEM images of HCQ NPs showed a spherical shape with a smaller particle size (Fig. 1A). The average hydrodynamic diameter of the HCQ NPs was approximately 55.36 ± 0.27 nm (Fig. 1B). Since solid tumors have enhanced permeability and retention effects, the nanosized diameters of HCQ NPs likely promote their accumulation at tumor sites. The zeta potential of HCQ NPs was affected by pH (Fig. 1C). The negative charge of HCQ NPs was weakened in acidic conditions compared to alkaline or neutral conditions. Carboxyl groups present in HA become ionized under weakly alkaline or near-neutral conditions compared to acidic conditions. XPS revealed the presence of Cu2+ in HCQ NPs. The high-resolution XPS spectrum of Cu2+ suggests that Cu existed primarily in the form of Cu2+ (Cu 2p 3/2 peak at 934.58 eV and Cu 2p 1/2 peak at 953.88 eV) with altered satellite peaks (940.48 eV and 943.58 eV) (Fig. 1D). After chelation with Cu2+, the UV−vis absorption spectrum of the characteristic peak of Qu at 254 and 374 nm was changed to 296 nm (Fig. 2A). Similarly, FTIR spectroscopy revealed that complex formation of Qu and Cu2+ substantially changed the wavenumber of the 4-position C = O characteristic absorption peak (Fig. 2B). The C = O vibration frequency of the carbonyl group shifted to the lower wavenumber direction (1612.53 and 1599.90 cm−1), corresponding to the formation of a coordinated bond between oxygen ions present on the carbonyl group and the metallic ions (Tan, Wang & Zhu, 2009). This bond deviated the electron density of the carboxyl group from the geometric center to a low-frequency shift, indicating that the 4-position carbonyl oxygen participates in the coordinated reaction. The stoichiometric ratio of Qu to Cu2+ was determined to be 1:2 based on infrared and ICP-MS analyses. The ICP-MS results showed 133 µg of Cu ions per milliliter of HCQ NPs. Based on the proportion of Qu and Cu2+ (1:2), the total drug loading of Qu and Cu2+ in the HCQ NPs for therapeutic purposes was calculated to be 9%. The drug loading capacity of HCQ NPs was similar to that of most nanodrug loading systems (~10%) (Li, Liao & Du, 2018). The molecular weight of HA is significantly greater than the Qu-Cu2+ complex. Therefore, introducing a large amount of HA (91%) leads to the formation of a Qu-Cu2+ complex nanodrug.

Figure 1 Characterizations of HCQ NPs.

(A) TEM images of HCQ NPs; (B) Size distribution of HCQ NPs; (C) Zeta potential under different pH conditions of HCQ NPs; (D) XPS analysis of HCQ NPs. Data are presented as mean ± SD (n = 3). **P < 0.01.

Figure 2 Characterizations of HCQ NPs properties.

(A) UV analysis; (B) FT-IR analysis; (C) Time stability, and (D) Temperature stability. Data are presented as mean ± SD (n = 3). **P < 0.01, ***P < 0.001, ****P < 0.0001.

The temperature stability test demonstrated that the HCQ-NPs were stable at room temperature (25 °C) and higher temperatures (up to 50 °C). However, at a low temperature (4 °C), HCQ NPs flocculated due to electrostatic interactions between HA and the Qu-Cu2+ complex, which was significantly influenced by temperature (Fig. 2D). The time stability test demonstrated that HCQ NPs were stable for 7 days at room temperature (Fig. 2C), which was due to the good water solubility of HA (Hsiao et al., 2015).

ROS product stability of HCQ NPs

Hydroxyl radical •OH is a potent oxidant that can react with various compounds, leading to their degradation (Buxton et al., 1988). In a biological environment, •OH can affect most cellular biomolecules present in proteins, amino acids, and lipids (Bogdan, Zarzyńska & Pławińska-Czarnak, 2015). The capacity of the HCQ NPs to produce •OH was investigated using an MB degradation assay. The MB content did not change in the HCQ NPs samples, but it was slightly decreased after H2O2 addition, which led to the generation of •OH by a Fenton-like reaction (Fig. 3A).

Figure 3 (A) Extracellular chemodynamic activity of HCQ NPs (A, PBS; B, HCQ; C, HCQ+H2O2 2 nM; D, HCQ+H2O2 4 nM; E, HCQ+H2O2 6 nM); (B) Fluorescence quantitative results of (C); (C) MDA-MB-231 cells incubated with PBS, HCQ, and HCQ+H2O2 after 24 h.

ROS production was determined using DCFH-DA after irradiation. Fluorescence was observed using a fluorescent microscope. Data are presented as mean ± SD (n = 3). **P < 0.01, ***P < 0.001.

The production of ROS in the MDA-MB-231 cellular environment was further confirmed using the fluorescence probe DCFH-DA. DCFH-DA is readily oxidized by ROS, emitting green fluorescence (Rastogi et al., 2010). The hypoxia and antioxidant substances (such as GSH) present in the TME may limit the production of ROS (Shi et al., 2020). Therefore, PBS control indicated that ROS production was low in that cell line under standard culture conditions. For HCQ group, Qu of HCQ NPs can reduce the concentration of antioxidant substances by decreasing the expression level of Bcl-2 (Sethi et al., 2023). Therefore, the ROS level was relatively increased compared to the PBS control. Confocal laser scanning microscopy (CLSM) images showed considerably higher green fluorescence in the co-incubation groups of HCQ NPs and H2O2 treated cancer cells compared to the control groups (PBS and HCQ NPs alone) (Fig. 3C). These findings confirmed the efficient generation of considerable amounts of •OH. They indicated the promising potential of HCQ NPs for synergetic CDT cancer therapy. As an emerging noninvasive cancer treatment method, CDT can convert excessive amounts of endogenously generated H2O2 in the TME to toxic •OH through either Fe-mediated or Cu-mediated Fenton-like reactions (Bokare & Choi, 2014; Tang et al., 2019). The essence of a Fenton or Fenton-like reaction is that Fe2+ or Cu1+ reacts with H2O2 to produce highly toxic •OH; however, Fe2+ or Cu1+ radicals are unstable and easily oxidized to Fe3+ or Cu2+. Since Fe3+ or Cu2+ containing reagents can reduce to Fe2+ or Cu1+ under appropriate conditions (Grinhut et al., 2011), they are preferable for Fenton reactions. Initially, Qu coordinated with Cu ions and reduced Cu2+ to Cu1+, which participated in the Fenton-like reaction. The formation and cleavage of the coordinated bond between 3-OH and Cu2+ in the Qu structure is a reversible reaction (Sun et al., 2020). When it breaks, Cu2+ reduces to Cu1+, which participates in the Fenton-like reaction, thereby further promoting the breakage of the coordinated bonds. Therefore, increasing the H2O2 concentration outside the cells increases the production of •OH (Tan, Wang & Zhu, 2009). In addition, there is a higher concentration of glutathione in tumor cells compared to normal tissue cells, which results in strong reduction of Cu2+ to Cu1+ (Liu et al., 2018). Therefore, HCQ NPs can efficiently generate ROS through Fenton-like reactions both inside and outside the cell (Figs. 3A–3C), providing evidence to support the use of HCQ NPs for CDT.

Cellular uptake of HCQ NPs

The cellular uptake of HCQ NPs by MDA-MB-231 cells was investigated using FCM. A higher mean fluorescence intensity indicated a greater uptake of HCQ NPs with longer incubation periods (0.5, 2, 4, 12, and 24 h), indicating a time-dependent uptake of HCQ NPs by MDA-MB-231 cells (Fig. 4A). These results also indicated that HCQ NPs could be rapidly taken up by tumor cells, and HCQ NPs could be rapidly distributed into tumor cells to exert synergistic antitumor effects.

Figure 4 (A) Flow cytometric results of HCQ NPs at different incubation times; (B) CCK8 assay results of LO2 cells. Survival ratio of cells after treatment with PBS and HCQ NPs at different concentrations (5, 10, 15, and 20 μL represent Cu:Qu = 100 μM:50 μM, 190 μM:95 μM, 274 μM:137 μM, and 350 μM:175 μM); (C) CCK8 assay results of MDA-MB 231 cells. Survival ratio of cells after treatment with PBS, HCQ NPs and HCQ NPs+H2O2 at different concentrations (5, 10, 15, and 20 μL represent Cu:Qu = 100 μM:50 μM, 190 μM:95 μM, 274 μM:137 μM, and 350 μM:175 μM); (D) Calcein AM and PI co-stained images of MDA-MB-231 cells incubated with different treatment groups.

Data are presented as mean ± SD (n = 3). *P < 0.05, **P < 0.01, ***P < 0.001.

Cytotoxicity and synergism assay

Considering the excellent capacity of HCQ NPs to produce •OH, the in vitro anticancer effect was further determined using the CCK-8 assay. LO2 cells were cultured with various concentrations of HCQ NPs (Cu:Qu = 100 μM:50 μM, 190 μM:95 μM, 274 μM:137 μM, and 350 μM:175 μM). The HCQ NPs showed negligible adverse effects on LO2 cell viability, demonstrating good cytocompatibility (Fig. 4B). Furthermore, the viability of MDA-MB-231 cells declined with increasing concentrations of HCQ NPs. A significant decline in cell viability was observed in the presence of H2O2 (simulated TME), indicating synergistically strengthened anticancer effects (Fig. 4C). To confirm further and visualize the cell death induced by chemotherapy and CDT, living and dying cells with respective green fluorescence and red fluorescence was observed using CLSM. Compared to the control group (PBS-treated MDA-MB-231 cells), some of the MDA-MB-231 cells were damaged following HCQ NPs treatment. Still, the majority of cells were damaged when simultaneously treated with H2O2 and HCQ NPs (Fig. 4D). These results further support the chemotherapeutic and ROS effects of HCQ NPs for synergistic tumor therapy.

There is a significant limitation in this study that should be noted. The study lacked animal experiments to investigate the antitumor effect of the HCQ NPs in vivo. In the future, we will conduct animal experiments to verify the synergistic and antitumor effects of the HCQ NPs.

Conclusions

This is the first study to report the synthesis of self-assembled flavonoid-coordinated polymer nanoparticles for chemotherapeutic applications. A Qu modification was used to introduce Cu2+ (as a linker) and HA to form a Qu-Cu2+ targeted nanodrug. In vitro studies demonstrated that HCQ NPs had excellent biocompatibility and synergistic antitumor effects. This study provides a novel method for applying traditional Chinese medicine extracts by forming similar coordinated polymer nanoparticles.

Supplemental Information

Supplemental Information 1 HCQ raw data.

Click here for additional data file.

Additional Information and Declarations

Competing Interests

Author Contributions

Data Availability

The authors declare that they have no competing interests.

Hanxun Yue performed the experiments, analyzed the data, prepared figures and/or tables, authored or reviewed drafts of the article, and approved the final draft.

Xuan Zhao performed the experiments, analyzed the data, prepared figures and/or tables, authored or reviewed drafts of the article, and approved the final draft.

Qin Yong performed the experiments, prepared figures and/or tables, and approved the final draft.

Min Shi performed the experiments, prepared figures and/or tables, and approved the final draft.

Xiaofeng Jiang performed the experiments, prepared figures and/or tables, and approved the final draft.

Yating Zhang performed the experiments, prepared figures and/or tables, and approved the final draft.

Xian Yu conceived and designed the experiments, analyzed the data, authored or reviewed drafts of the article, and approved the final draft.

The following information was supplied regarding data availability:

The raw measurements are available in the Supplemental File.

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
