# Peer review of "Self-assembly of Hyaluronic Acid-Cu-Quercetin flavonoid nanoparticles: synergistic chemotherapy to target tumors"

_PeerJ, doi:10.7717/peerj.15942_

## Round 0.1 · original submission · Major Revisions

Please attend to the reviewer's comments and provide a thoroughly revised manuscript version.

Reviewer 1 ·

Basic reporting

I thoroughly reviewed the manuscript, which deals with the production of nanoparticles made from hyaluronic acid and loaded with copper-functionalized quercetin to target cancer cells. I consider that the topic is relevant in the field of health and that the manuscript meets the basic requirements. There is consistency between all manuscript sections. Although I am not a native English speaker, I find the manuscript well-written.

Experimental design

An experimental design as such is not mentioned in the methodology. However, where necessary, mention of the number of repetitions and the respective standard deviations are reported. In my opinion, the work was well conducted and the results are robust. The methods are described in sufficient detail.
However, there are some details that I consider should be considered and I mention them in the next sections.

Validity of the findings

The study can be easily replicated since sufficient details of the methodology are given. The results are consistent with what was proposed in the introduction and the conclusion is supported by them.

Additional comments

Many of the studies on cancer are carried out in vitro, under conditions that are intended to be at least very similar to the physiological ones. Many of these results are very promising, but they do not go beyond that because, in real-life situations, they are not reproducible or have never been tested in humans. How could the authors argue that their study is not one of the many and only come off as a potentially good thing?

In the background, it is argued that Qu's poor water solubility is one of the reasons that restrict its clinical applications. In the present study, the water solubility of the nanoparticles was not analyzed. Could you please elaborate more on this topic?

Discussions in the characterization results (L 200-221) are not supported by references.
In line 201 of the text, it is said that the hydrodynamic radius was 78.6 nm. How was this result reached since, in Fig. 1c, the mean of the peak is approximately 35 nm? Furthermore, when comparing the sizes obtained by DLS with those obtained by TEM, the most appropriate would be to express the distribution obtained by DLS as number distribution and not as volume distribution. I suggest converting the data to do the comparison, as well as the following article:
Nobbmann and Narfesis. 2009. Light scattering and nanoparticles.
Materials Today 12(5):52-54. DOI: 10.1016/S1369-7021(09)70164-6

Also, the following question arises: What would be the route of administration of the nanoparticles loaded with the active compound? Explain the journey these nanoparticles would make to reach the target tumor and how they would potentially solve the problems that membrane barriers and different pH conditions imply, etc.

Reviewer 2 ·

Basic reporting

Yue et al, synthesized novel nanoparticles by self-assembling Quercetin, a flavonoid known to have anticancer properties, metal ion (cu), and hyaluronic acid (HA). Quercetin is known for its anticancer properties. However, its low solubility, bioavailability, and lack of targeted delivery are major challenges to using it for cancer treatment. In this manuscript, the authors synthesized novel nanoparticles (HCQ NPs) for targeted delivery with improved biocompatibility, and they tested efficacy in two cancer cell lines. Yue et al. found that the HCQ NPs exhibited better biocompatibility and showed toxicity against one of the two cancer cell lines tested. Overall, this is a good study. However, it needs to be improved significantly. I have outlined the few points which I am most concerned about. I am sure that the authors can address most of my concerns.

Major comments
1) The Qu-Cu NPs have been tested previously in cancer cell lines. In this MS, the authors made one more modification by combining the HA. But it needs to be clarified if HCQ NPs are any better than Qu-Cu. Is it possible to do an experiment to compare between Qu-Cu and HCQ Nps?
2) The author tested the HCQ only in two cell lines and showed toxicity against only one cell line. Can the author test HCQ NPs on a few more cell lines to see if they are effective against any other cell lines?
3) In Fig 1a, the authors claimed that they were able to synthesize uniformly sized NPs.
However, many different sizes can be seen in the TEM shown. This needs to be clarified. The author could report the size in mean SD if there were different size particles.
4) Can the authors quantify the data in Fig 3b?
5) None of the data has been statistically analyzed. Please provide statistical analysis for the quantification data.

Minor comments
6) In Fig 1d, please define the different color lines and mark the altered satellite peak as claimed in line 209.
7) In Fig 2c, 2d, please provide the number of particles measured.
8) Please define X-axis in Fig 4b and 4c. Also, provide the statistical analysis.
9) The scale bars are missing in IF images in Fid 4d.

Experimental design

Comparison with other Nanoparticles needs to be included. The MS will be much better if the authors can provide a comparison of HCQ NPs with Qu-Cu.

Validity of the findings

The findings are valid. But quantification and statistical analysis should be provided.

Reviewer 3 ·

Basic reporting

The manuscript "Self-assembly of hyaluronic acid-Cu-quercetin flavonoid nanoparticles: synergistic chemotherapy to target tumors" describes the synthesis of bioactive multifunctional, flavonoid nanoparticles (HCQ NPs) containing HA, CuCl2, and Qu for potential anticancer applications. In general, the manuscript is well written. The literature references provide sufficient field background. The study provides valuable scientific information, however a major revision must be done in order to continue with its publication within the journal.

Experimental design

The manuscript describes the characterization of synthesized nanoparticles, as well as their pharmacological potential by in vitro and cellular assays. The methodological approach is appropriate and is clearly described with sufficient detail, and the results are within the scope of the journal. However, it would be convenient to compare the bioactivity ot quercetin (Qu) to Qu and Cu2+, and then to HCQ NPs, and finally address this information to the bioactivity of the total drug loading of Qu and Cu2+ (about the 9 %).

Validity of the findings

The results described within the manuscript are of scientific relevance. The characterization of nanoparticles is solid, however the results concerning the in vitro experiments would be more supported if quantitative information about Mean Fluorescence Intensity be included in addition to micrographs obtained by confocal fluorescence microscopy analysis (Fiugre 3 and 4).
Why the authors did not include H2O2 alone control (Figure 3). Interesting information about basal redox state of cellular model and oxidative stress induced with H2O2 could be discussed.

Additional comments

Minor revision:
- Line 69-71, page 6: "In vivo and in vitro experiments have shown that Qu exerts its antitumor effects by altering the cell cycle, inhibiting cell proliferation, promoting apoptosis, and inhibiting angiogenesis..... ", Please indicate how cell cycle is altered (cell cycle arrest at which phase), and address this bioactive potential to redox state imbalances.
- Line 180 and 182, page 9, please correct "natural medicines" to naturally bioactive compounds.
- Line 223 to 224, page 10, "Therefore, introducing a large amount of HA leads to the formation of a Qu-Cu2+ complex nanodrug", please indicate the proportion.
-Page 17, figure 1 (c), please correct Y-axis, change "zata potential" to "zeta potential".

---

## Round 0.2 · Minor Revisions

I kindly request that you draft a rebuttal letter that addresses specific line numbers in the revised manuscript. Additionally, I would like to bring to your attention that the comment regarding Figure 1b is still relevant. The reference to Figure 1c was made in error, and it was clear from the context that the reviewer was referring to the Figure where the hydrodynamic diameter is reported. I would appreciate it if you could take this into consideration.

Reviewer 1 ·

Basic reporting

No comment

Experimental design

No comment

Validity of the findings

No comment

Additional comments

Dear authors,
Thank you for taking the observations into account, however it would be very useful if you would indicate in which lines of the revised manuscript the changes were made, since I cannot find them except for what is related to the hydrodynamic radius. It is not enough to offer an explanation to the reviewers and that this is not reflected in the manuscript, since the observations are to improve it. By the way, I apologize for a mistake I made when I made the observation regarding the hydrodynamic radius, I wanted to refer to Figure 1b and not Figure 1c. Therefore, the question remains: why do you establish that the average hydrodynamic radius is 78.6 nm if the logarithmic scale of Figure 1b shows that it is approximately 40 nm? I also think that it would be better to report it as diameter and not as radius since the former is more usual.

Reviewer 2 ·

Basic reporting

Yue et al., synthesized novel nanoparticles by self-assembling Quercetin, a flavonoid known to have anticancer properties, metal ion (cu), and hyaluronic acid (HA). Quercetin is known for its anticancer properties. However, its low solubility, bioavailability, and lack of targeted delivery are significant challenges to using it for cancer treatment. In this manuscript, the authors synthesized novel nanoparticles (HCQ NPs) for targeted delivery with improved biocompatibility, and they tested efficacy in two cancer cell lines. Yue et al. found that the HCQ NPs exhibited better biocompatibility and showed toxicity against one of the two cancer cell lines tested.

In this revised version of the MS, the authors have addressed most of the comments. I thank you, authors, for your effort to answer most of the comments. I think this MS can be accepted for publication.

Experimental design

This study is original and falls under the aims and scope of the journal. Methods are described with sufficient detail, and the experimental design is robust.

Validity of the findings

All underlying data have been provided; they are robust. The statistical analyses have been provided where applicable. The conclusion is well-stated.

Reviewer 3 ·

Basic reporting

The manuscript has been improved by the authors, according to reviewers' comments. However several issues must be clarified.
Language must be revised.
Please change HCQ NP characterizations or characterizations HCQ NP to HCQ NP characterization or characterization HCQ NP throughout the manuscript.

Experimental design

The drug loading content of the two drugs (Qu and Cu) in the HCQ NPs was calculated, therefore, why not the authors express the concentration of Qu and Cu in the synthesized HCQ NPs (figure 4) as they described in Line 280-282: “…cells were cultured with various concentrations of HCQ NPs (Cu:Qu = 100 μM:50 μM, 190 μM:95 μM, 274 μM:137 μM, and 350 μM:175 μM)…”, instead of illustrating volumes? Please correct.
Lines 144-145: “…PBS, HCQ, and HCQ + H2O2 groups were respectively incubated with 55 μL PBS, 50 μL HCQ NPs + 5 μL PBS, and 50 μL HCQ NPs + 5 μL H2O2 for 4 h…” Please indicate if HCQ NPs were dissolved in PBS, because the information above suggests that PBS was not the dissolvent for synthesized NPs but for H2O2, and therefore the strategy is not clear. Please indicate the origin of H2O2 reagent and its original concentration, as well as final concentration of H2O2 in cellular assays.
Cancer cell lines are used in this study as a model, therefore basal redox state of cancer cell lines would indicate the basal metabolic activity and cellular behavior in culture. PBS control dissolvent in DCFH-DA assays indicates that ROS production is low in that cell line under normal culture conditions, please discuss this and how HCQ increases ROS and quantify the differences between groups, (MFI increasing ratio), as well as HCQ +H2O2. Please explain why the authors responded this: “Our original experiments were conducted under the simulated tumor microenvironment, which is characterized by a high concentration (relevant studies showed that the concentration could reach 100 uM or even higher1). Therefore, no H2O2 control group was included”.
Line 207-210, please describe the average hydrodynamic diameter of the HCQ NPs as mean ± SD of at least three independent determinations, since figure 1 depicts that distribution of size of synthesized particles could be from 20 to 90, therefore... the average size of NPs, according to the Gaussian distribution of the peak in Fig 1B, is not 78 nm.

Validity of the findings

Line 286-288: “…To further confirm and visualize the cell apoptosis induced by chemotherapy and CDT, living and dying cells with respective green fluorescence and red fluorescence were observed using CLSM…”. By using calcein and PI it is not possible to determine apoptosis activation, since calcein-PI is a common double fluorescence staining to discern viable and dead cells according to intracellular esterase activity (calcein metabolic activity) and cell membrane integrity (PI). The mechanisms behind cell death can not be analyzed by using this staining. Immunochemistry assays for Annexin V or caspase activation pathways could be appropriated to discuss apoptosis.

---

## Round 0.3 · Minor Revisions

Thank you for addressing the suggestions made by the reviewers. I have included some suggestions regarding style and typos that could improve the manuscript. Please consider them at your convenience.

---

## Round 0.4 · accepted · Accept

Thank you for addressing all of the requested revisions and corrections. Your manuscript has now been accepted by PeerJ.